# VideoFlow: A Framework for Building Visual Analysis Pipelines

## Abstract

The past years have witnessed an explosion of deep learning frameworks like PyTorch and TensorFlow since the success of deep neural networks. These frameworks have significantly facilitated algorithm development in multimedia research and production. However, how to easily and efficiently build an end-to-end visual analysis pipeline with these algorithms is still an open issue. In most cases, developers have to spend a huge amount of time tackling data input and output, optimizing computation efficiency, or even debugging exhausting memory leaks together with algorithm development. VideoFlow aims to overcome these challenges by providing a flexible, efficient, extensible, and secure visual analysis framework for both the academia and industry. With VideoFlow, developers can focus on the improvement of algorithms themselves, as well as the construction of a complete visual analysis workflow. VideoFlow has been incubated in the practices of smart city innovation for more than three years. It has been widely used in tens of intelligent visual analysis systems. VideoFlow will be open-sourced at `https://github.com/xxx/videoflow`.

## 1 Introduction

The success of computer vision techniques is spawning intelligent visual analysis systems in real applications. Rather than serving individual models, these systems are often powered by a workflow of image/video decoding, several serial or parallel algorithm processing stages, as well as sinking analysis results. The varied visual analysis requirements in different real scenarios put forward a high demand on a framework for fast algorithm development, flexible pipeline construction, efficient workflow execution, as well as secure model protection.

There exist some frameworks approaching some of the above mentioned targets, like DeepStream (Purandare, 2018) and MediaPipe (Lugaresi et al., 2019). DeepStream is on top of GStreamer (GSTREAMER, 1999), which primarily targets audio/video media editing rather than analysis. MediaPipe can be used to build prototypes to polished cross-platform applications and measure performance. Though it is flexible and extensible on calculators, efficiency, model security, and extension on more aspects are expected by real online services in industry.

In this paper, we present VideoFlow, to meet the visual analysis requirements for both algorithm development and deployment in real systems with the following highlights.

**Flexibility**. VideoFlow is designed around stateful *Computation Graph* and stateless *Resource*. Computation graph abstracts the visual processing workflow into a stateful directed acyclic graph. Developers can focus on the implementation of processing units (graph nodes) and the construction of the whole workflow. *Resource* is a stateless shared computation module of computation graphs. The most typical resource is deep learning model inference. Resource decouples the stateless visual processing components from the whole complicated visual analysis pipeline, helping developers focus on the optimization of these computation or Input/Output(IO) intensive implementation.

**Efficiency**. VideoFlow is designed for better efficiency from four levels. (1) Resource-level: resources can aggregate the scattered computation requests from computation graph instances into intensive processing for better efficiency. (2) Video-level: all videos are analyzed in parallel in a shared execution engine. (3) Frame-level: video frames are parallelized on operations which are irrelevant to frame orders. (4) Operator-level: visual analysis is a multi-branch pipeline in most

cases. The different branches and different operators of each branch without sequential dependency are analyzed in parallel.

**Extensibility**. VideoFlow is designed from the beginning to be as modular as possible to allow easy extension to almost all its components. It can be extended to different hardware devices like Graphic Processing Units(GPU), Neural Processing Unit (NPU), etc. It can be hosted on either x86 or ARM platforms. Developers can customize their own implementations with VideoFlow as a dependent library. The extended implementations can be registered back to VideoFlow as plugins at runtime.

**Security.** Model protection is an important problem in industry. VideoFlow encodes model files into encrypted binary codes as part of the compiled library. The secret key can be obscured into the same library, or exported to a separate key management service (KMS). At runtime, VideoFlow decrypts the models and verifies authorization from a remote service periodically.

VideoFlow has been incubated in the practices of the smart city innovation for more than three years. It is designed for computer vision practitioners, including engineers, researchers, students, and software developers. The targets of VideoFlow include: 1) free developers from the exhausting data loading/sinking, parallel programming and debugging to the optimization of algorithms; 2) enable easy extension of video decoding, deep model inference and algorithm implementation; 3) provide highly efficient framework for large scale visual processing in industry rather than just experimental prototypes. 4) protect the intellectual property of models and algorithms to make sure that they can only work with authorization.

## 2    RELATED WORK

### 2.1    DEEP LEARNING FRAMEWORKS

Almost all existing deep learning frameworks like Caffe (Jia et al., 2014), TensorFlow (Abadi et al., 2016), PyTorch (Paszke et al., 2017), MXNet (Chen et al., 2015) describe networks in directed graphs or even dynamic graphs. VideoFlow draws lessons from this successful design for visual analysis. The difference is that the basic units in deep networks are low level operations like convolutions, compared to higher level processing like object tracking in VideoFlow. The data transferred between operators in VideoFlow is also much more complex than the *Tensor* in deep learning.

As to model inference, there are some specially optimized engines , like TensorRT (Vanholder, 2016) and MKL-DNN/oneAPI (Intel) by hardware manufactures. In the open source community, developers put forward TVM for easy extension to different hardware for more effective inference (Chen et al., 2017). On top of these engines, there are some serving platforms for individual models rather workflow construction, like tensorflow serving (Google, 2016), NVIDIA Triton Inference Server (Goodwin & Jeong, 2019). VideoFlow integrates these inference engines as Resources with their C++ interfaces.

### 2.2    VISUAL ANALYSIS FRAMEWORKS

The recent has witnessed some visual analysis frameworks. Nvidia launches the DeepStream project in the early days for video analysis on GPU (Purandare, 2018). It is oriented as well as optimized on GPU and TensorRT, regardless of the bustling heterogeneous hardware devices. Besides, it is built on top of GStreamer (GSTREAMER, 1999), which primarily targets audio/video media editing rather than analysis, limiting its flexibility and extensibility. The *gst-video-analytics* project (Intel, 2019) is also built on top of GStreamer(Deuermeyer & Andrey). Google proposed MediaPipe by building graphs for arbitrary streaming data processing with a computation graph as well (Lugaresi et al., 2019). MediaPipe can be used to build prototypes to polished cross-platform applications and measure performance. Though it is flexible and extensible on calculators, real online visual analysis expects extension on more aspects, more efficiency optimization, and model security protection. Compared to MediaPipe, VideoFlow features these advantages for better application in both academia and industry. Another framework also named Videoflow (de Armas, 2019) is designed to facilitate easy and quick definition of computer vision stream processing pipelines. However, it is just a prototype experimental platform, with limitations on extensibility, efficiency, and security.

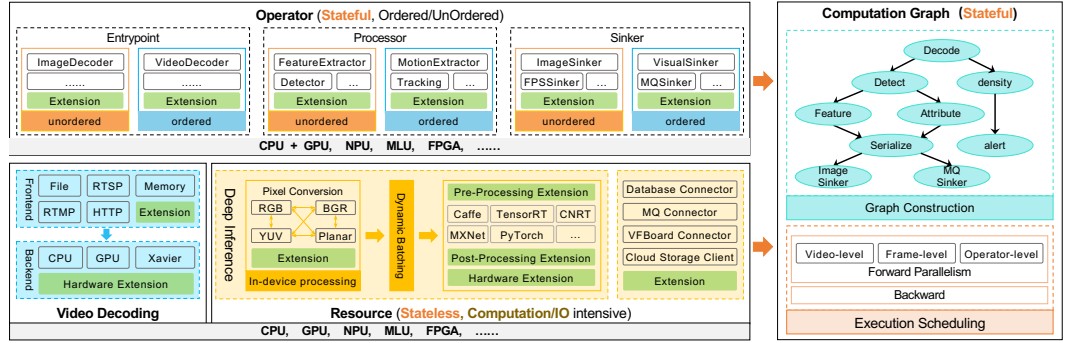

Figure 1: Framework of VideoFlow.

# 3 ARCHITECTURE

VideoFlow is oriented around stateful *Computation Graph* and stateless *Resource* with a well-optimized execution engine. Computation graph is a directed acyclic graph describing the whole workflow of video or image analysis. As the two main components of a graph, *Node* and *Edge* denote visual processing operators and data flow between operators, respectively. *Resource* is shared for graph irrelevant computation. The architecture is shown in Figure 1.

## 3.1 OPERATOR

Operator is the basic unit of visual analysis workflow. An operator depends on the outputs of its parent operators. Its own outputs can be consumed by arbitrary number of child operators. According to the number of inputs an outputs, operators are categorized as follows:

- **Entrypoint**: operators that have zero inputs. This is the start of a computation graph. Each graph can have only one entrypoint.

- **Processor**: operators that have at least one input and at least one output. Processors occupy most of the workflow of visual analysis. It's also the main kind of operator with the highest demand on easy extension.

- **Sinker**: operators that have zero outputs. This is the end of a computation graph. A graph can have multiple sinkers.

## 3.2 DATA FLOW

Data flow is the edge connection between two operators (*nodes*). An operator may generate several number of data with different types for its child nodes. Data flow is a collection of arbitrary number of data pointers of arbitrary type (vector<void*> in our C++ implementation) in VideoFlow. VideoFlow guarantees that the incoming data pointers are always safe to be read. Developers do not need to care how many other operators are also consuming the data, or whether the data should be released during the workflow.

## 3.3 RESOURCE

Resource is the stateless computation unit shared by graphs. The most representative resource is deep model inference. Resource is abstracted due to three main reasons. Firstly, many operations like deep model inference and data sinking to databases have their own independent semantics. They are irrelevant to whether it is used for video or image processing, which step of the whole pipeline invokes the operation, or how the outputs will be post-processed. Secondly, these operations are often computation or IO intensive. Leaving them in the operators will incur bottlenecks on CPU, memory or network bandwidth due to large amount of resource competition. Gathering the scattered but shared requests from different graphs for uniform processing proves to be a good practice to improve efficiency. Thirdly, resource can be improved without affecting the visual analysis logic.

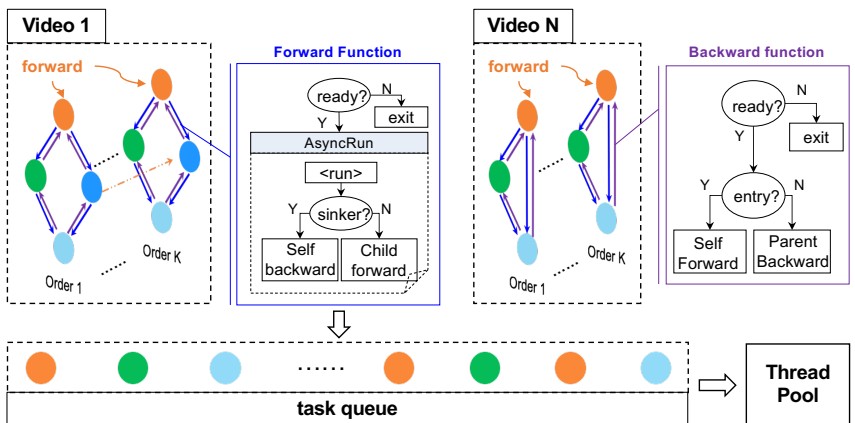

Figure 2: Execution Scheduling of VideoFlow.

For example, we can accelerate the inference speed of a PyTorch model by switching to a TensorRT worker. We can change to a more efficient database connector for more real-time data sinking. Without the abstraction of these resources, all affected operators have to be re-implemented to earn the benefits.

### 3.4 GRAPH CONSTRUCTION

Computation graph is described in json format with the following fields.

**"resource"** describes the resources that will be used by operators. Each resource should have two fields: "type" to create the correct class instance and "params" to specify the resource configurations.

**"ops"** describes the operators that will be used to construct computation graphs. Operators can be used multiple times by different graphs. As the same to resource, each operator should have two fields: "type" and "params".

**"graph"** is the place to define computation graphs. Each graph definition is a json dictionary of key-value pairs. Key is the operator name. Its value is a list of operator names as its child nodes.

**"subgraphs"[optional]** is used to re-use resources, operators and graphs from other graph configuration files.

**"libs"[optional]** specifies external dynamic libraries that should be loaded by VideoFlow, especially the extended libraries in Figure 3.

**"config"[optional]** is for global settings, currently including number of parallel image processing threads and number of frames for parallel video processing.

An example file is provided in the supplementary material to show the person reidentification workflow (Section 5).

### 3.5 EXECUTION SCHEDULING

With the graph defined and constructed, execution scheduling determines which operator should be calculated. In real cases, there can be multiple computation graph instances running in parallel, each with either shared or different structures. Figure 2 shows the execution scheduling of these graphs. Each graph has several replicas, with each replica called as an order. Video frames are actually processed in these graph replicas/orders. The replicas are processed in parallel for frame-level parallelism. Each order starts from the Forward function of the entrypoint node.

**Forward.** As Figure 2 shows, the forward function first checks if the current operator is ready to be executed. The readiness checking includes: 1) all parents of the current node have finished on this order. 2) the previous order of the current node has been executed if the current node is an ordered operator. If ready, the forward function puts its own processing function into the task queue

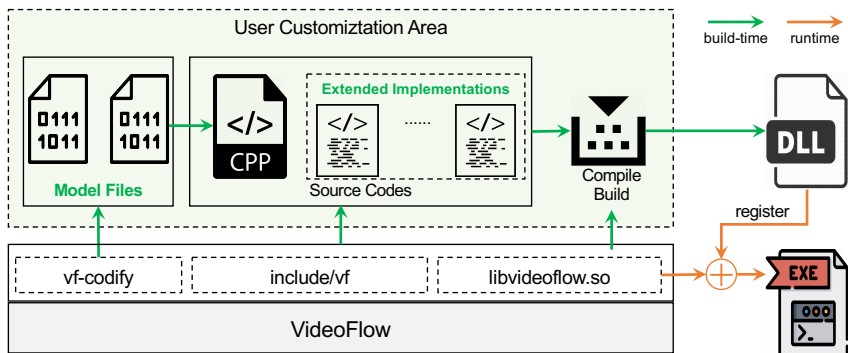

Figure 3: Illustration of extension. Developers depend on VideoFlow to build their library at build-time. At runtime, the generated library is loaded by VideoFlow dynamically. Extended operators, resources and other modules are registered back to VideoFlow.

of the execution engine, waiting to be executed. The processing function first finishes the internal processing logic of the operator. After that, it calls the `Forward` function of its following operators. If it is the leaf operator, it calls the `Backward` function of its own. *Forward* of entrypoints is specially implemented with a separate thread retrieving and dispatching data to idle orders.

**Backward** is the process to reset the node to be ready to process latter frames. The backward function first checks if all its children have been reset. If so, it resets its forward status. Then it continues to call the `Backward` of all its parents.

**Execution Engine.** The processing functions of operators are put into a task queue of the execution engine. All processing units share the same interface. The execution engine does not know which order of which graph a processing function comes from. All orders and all graphs are executed concurrently once they are put into the queue. Inside the engine there is a thread pool, with all threads fetching and executing tasks from the queue.

### 3.6 DEEP MODEL INFERENCE

Deep model inference is the most typical computation intensive resource in visual analysis. The built-in implementation covers deep learning frameworks (Caffe, PyTorch, MXNet)[1] and acceleration toolkits (TensorRT, TVM, etc.). Developers can customize their own workers easily. VideoFlow adopts dynamic batching and in-device processing to fully utilize the capability of heterogeneous hardware devices. On GPU, it supports multi-stream for better performance.

**Dynamic Batching.** Heterogeneous hardware devices often need to work in the batch mode for full utilization. A single call of a batch of data is much more efficient than multiple calls of single data. In VideoFlow, the input of deep learning inference is a batch of `DLTask`, which is a structure defined for deep learning input and output of a single task. VideoFlow defines a thread-safe queue to collect the scattered `DLTask` from operators to form a `DLTaskBatch`. Deep inference workers request `DLTaskBatch` from the queue with a timeout mechanism. The timeout is essential since videos, frames, and operators are not strictly time aligned. The timeout can try to collect as many tasks as possible in the limited period, while keeping the processing latency limited at the same time.

**In-device processing.** Data transfer across host and device memory is cost expensive and time consuming. In VideoFlow, frames are defined with hardware contexts. It provides hardware-specific image operations like resizing, cropping, pixel format conversion, mean subtraction, scaling, etc. All these operations including dynamic batching are conducted on their most appropriate hardware devices according to their hardware contexts. GPU-decoded frames are pre-processed and analyzed on the same GPU card. CPU-decoded frames are pre-processed on CPU, but analyzed on device with the lowest expected latency. The in-device processing lowers the system cost significantly, especially the CPU cost, as will be verified on the Section 5.

---

[1]Currently TensorFlow is partially supported due to difficulties in integrating its C++ interface and library.

### 3.7 EXTENSION OF VIDEOFLOW

Library extension is often intrusive. Developers have to write and compile their code together within the libraries. Though it works, the drawbacks are: 1) The total compilation time is getting longer with the library becomes more and more complex; 2) It is really hard to integrate extensions from different developers except that they share the same code repository. However, the different extension may come from different teams, with their concern for code and model protection.

VideoFlow provides a more convenient way for extension as Figure 3 shows. Developers can customize their own implementation and generate their own libraries. These libraries can have their own model protection or authorization. With the registration mechanism, VideoFlow will load libraries specified in the "libs" field of the graph configuration file at runtime. Throughout VideoFlow, registration is widely used for extension of all modules. By this way, VideoFlow enables coding separately, but working together.

### 3.8 OPERATOR EXTENSION

Operator interface is highly simplified for easy customization. All operators derive from the same base class `Op`. As the start of graphs, customization of entrypoints requires a little bit more care of data input and workflow interruption. These are highly dependent on how the graph will interact with outside callers. The built-in entrypoints have covered most cases of visual analysis. Except for entrypoints, there are only five steps to customize a new operator: *construction/destruction, initialization, auxiliary memory management, processing, and registration*. We provide a detailed example code of an object detection operator in the supplementary material.

**Construction/Destruction.** In construction function, developers should firstly make clear whether the current operator can be parallelized on the frame-level, which is called *ordered*. Frame sequence order independent processing should all be declared as unordered for better parallelism to boost the overall efficiency. The second is the number of parent operators and the input data type list. The last is the output type list of the current operator. Note that "*" is allowed during type specification for wildcard type. The output types of the parent operators will be checked to see whether they match the input types of child operators during graph construction.

**Initialization.** The `Init` function is used to initialize some settings after the graph has been constructed, but before the actual analysis begins.

**Auxiliary Memory Management.** Auxiliary memory is the frequently used temporal memory during processing. The life-cycle of auxiliary memory is the same with the graph. Auxiliary memory can be used as the output data. To be specific, developers need to override the `MallocResource` and `ReleaseResource` functions for auxiliary memory management.

**Processing.** This is the place to process the input data and generate the output data. The `run` function will be called again and again for visual analysis. All parallelism are optimized around this function, though developers do not need to care about the detailed mechanism. They only need to remember not to write shared memories (like class member variables or global variables) if the operator is declared as unordered for thread-safety, since there can be multiple threads executing the same *run* function on different frames. Note that the auxiliary memory will be allocated for each order of the graph. It is safe to operate on the auxiliary memory without thread-safety concern no matter the operator is ordered or not.

**Registration.** Registration is just a macro to register the operator to VideoFlow so that it can be constructed according to its name.

### 3.9 DECODER EXTENSION

Video decoding needs to tackle various video sources like online camera recording, web streaming, local files, or even non-standard video transfer protocols. Besides, it should make full use of the hardware decoding modules which are widely equipped with modern heterogeneous hardware devices. VideoFlow abstracts video decoding into frontend and backend. The frontend tackles various video sources. For backend, there can be different implementation on different hardware devices.

These backends register to `DecoderBackend` with their supported codecs, decoding capabilities, and priority.

To extend video decoding, developers can choose to implement a new video frontend or a new decoding backend. Interfaces of both the frontend and the backend are simple (`Open`, `Put`, `Get`, `Close`). VideoFlow makes sure that frontend developers do not need to care about decoding acceleration with hardware devices. Backend developers do not need to care about where the video data comes from or how to demux the data packets.

### 3.10 DEEP MODEL INFERENCE EXTENSION

VideoFlow provides built-in inference support for Caffe, MXNet, and PyTorch, as well as TensorRT and TVM (Chen et al., 2017). Nevertheless, there are still three main scenarios to extend deep model inference: new inference backend, output post-processing, and input pre-processing.

**Inference Backend** means to extend new deep learning or hardware acceleration frameworks. There are three functions to be overridden: `Init`, `FeedData`, and `process`. The `Init` function should determine the hardware context, parse the models, check model input and output, and allocate input/output memory buffers. `FeedData` is used to pre-process and feed a batch of data to the inference framework. `process` is a private function to invoke the inference after the input data is ready. The output is written back to `DLTask` in this step.

**Output Post-processing.** This is quite common for frameworks like TensorRT since there are still quite a large number of operators not supported. In many cases, developers can accelerate the backbone network with these acceleration toolkits. The left unsupported procedures can be implemented by overriding the `process` function of existing inference backends.

**Input Pre-processing.** VideoFlow provides scaling with mean subtraction and data copy as the built-in pre-processing methods for input image data. To customize a new pre-processing function, VideoFlow defines an empty template structure named `PreProcessFunc` with device type, pixel format, and pre-processing type as its template parameters. Developers just need to implement a new template specialization for their own pre-processing.

### 3.11 SECURITY

Model protection is a key challenge in deployment. VideoFlow provides a security mechanism as Figure 3 shows. *vf-codify* converts model files to encrypted source. The encryption key is obfuscated in the source codes to avoid library file parsing. In real production deployment, the key should be deposited into a Key-Management-Service (KMS) for higher security level.

At runtime, the security module tries to request authorization and decryption key from a remote service. If authorized successfully, it will decrypt the models into memory. Users may still have the concern if others can peep their models from the memory. This problem does exist in real cases. Attackers may dump the whole memory and steal the model parameters. A possible solution may be that hardware manufactures provide a safe memory region for model parameters, or part of model parameters. Anyway, security is an endless game of attack and defense. We welcome more open source contributions for better security.

## 4 TOOLS

VideoFlow provides a graph editor to help users write their computation graphs, a visualizer to visualize the real-time analysis results of algorithms, like the detected objects, object trajectories, as well as a profiler to show the running status of each video channel and deep learning models. These tools can significantly benefit users for fast pipeline construction and optimization. Detailed illustration of the tools is shown in the appendix.

## 5 PERSON REID APPLICATION EXAMPLE

In this section, we show an application example with a person reidentification (ReID) system. We test the performance benchmark of some measurable aspects of VideoFlow to verify its efficiency.

Table 1: Models used for Person ReID.

| Task | Model | Framework | Acceleration |
|------|-------|-----------|--------------|
| Person Detection | SSD (Liu et al., 2016) | Caffe[2] | TensorRT |
| Person Attribute | APR (Lin et al., 2019) | PyTorch[3] | TensorRT |
| Person Feature | Resnet50 (Guo et al., 2020) | MXNet[4] | TensorRT |

Table 2: Evaluation of Person ReID Pipeline.

| Optimization | | | | Resource Consumption | | | | Video Parallel | Average FPS |
|---|---|---|---|---|---|---|---|---|---|
| Frame Parallel | TensorRT | GPU Decode | In-device Processing | CPU | Memory | GPU | GPU Memory | | |
| 1 | | | | 95% | 9.0% | 30% | 12% | 1 | 17 |
| 4 | | | | 290% | 9.5% | 70% | 30% | 2 | 25 |
| 4 | ✓ | | | 1560% | 9.7% | 56% | 30% | 9 | 25 |
| 4 | ✓ | ✓ | | 1100% | 10.2% | 70% | 39% | 10 | 25 |
| 4 | ✓ | ✓ | ✓ | 800% | 10.2% | 70% | 39% | 10 | 25 |
| 8 | ✓ | ✓ | ✓ | 950% | 11.1% | 84% | 45% | 12 | 25 |

Person ReID is widely explored in recent computer vision community (Song et al., 2019; Yu et al., 2019; Sun et al., 2018; Wei et al., 2018; Zhao et al., 2017; Wei et al., 2017; Hermans et al., 2017). A typical person ReID video processing pipeline is shown in Figure 4 in the appendix. We sample once every 5 frames due to large content similarity. The decoded frames are processed with person detection and person tracking. After then, two branches extract the ReID feature and the person attributes, respectively. Sinkers include a `VisualSinker` for real-time visualization and a `FPSSinker` to calculate the processing speed. The pipeline is deployed on a cloud virtual machine with Intel Xeon E5-2682 CPU (16 cores), 64 GB memory, and 1 Nvidia T4 GPU card.

**Data.** We choose MOT16-07 video sequence from mot challenge (Milan et al., 2016). We convert its resolution to 1920x1080 and push it with FFmpeg in a loop as a rtsp video service (Schulzrinne et al., 1998). Different channels of VideoFlow pull the same video stream to mimic parallel video analysis.

**Models.** The three models are trained with Caffe, PyTorch, and MXNet to verify the wide support of deep learning frameworks, as shown in Table 1. These models will be accelerated with TensorRT for fast inference.

**Pipeline Benchmark.** Table 2 shows the efficiency evaluation of the whole pipeline. We choose some optimization aspects that can be quantified to verify the efficiency. Without frame parallel processing, even a single channel of video cannot be processed in real-time. Note that the workflow in this example is not that complicated. With four orders of frame parallelism, VideoFlow can process 2 video streams. TensorRT significantly boost the capability to 9 channels, as well as the CPU cost to almost 16 cores. GPU decoding reduces 5 cores of CPU consumption, with still 11 cores left mainly for person tracking. With in-device processing and larger frame parallelism, we can reduce about 1.5 cores of CPU consumption while improving concurrent real-time video processing to 12 channels.

## 6 CONCLUSION

In this paper, we present VideoFlow, a computation graph and resource based framework for visual analysis. We illustrate its superiority from the aspects of flexibility, efficiency, extensibility, and security. VideoFlow can help developers focus on algorithm improvement and the construction of visual analysis workflow. It is a carefully designed and implemented framework for the ease of visual analysis without bias to any hardware manufacturers, devices, platforms, or computer vision frameworks. It can be used in both academic and industrial scenarios.

VideoFlow has been widely used in intelligent visual analysis systems. We will open-source the project to welcome the contribution of the community for more implementation of operators, more adaptation of different hardware devices, as well as better optimization of the framework itself.

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

# A APPENDIX

The appendix describes the tools provided by VideoFlow for computation graphs construction, real-time analysis visualization, and operator performance profiling.

## A.1 GRAPH EDITOR

Graph editor aims to help users write their computation graphs. As shown in Figure 4, it is comprised of three main panels: operator list, graph editor, and graph visualizer. The operator list panel shows the list of all operators inside a library. Users can import several library metadata into this tool. Libraries can be switched in this panel. The graph editor panel supports smart editing of graphs with auto-completion, grammar checking and auto-formatting. The graph visualizer panel synchronizes the graph definition in the editor into a graphical view to help users review the whole processing workflow.

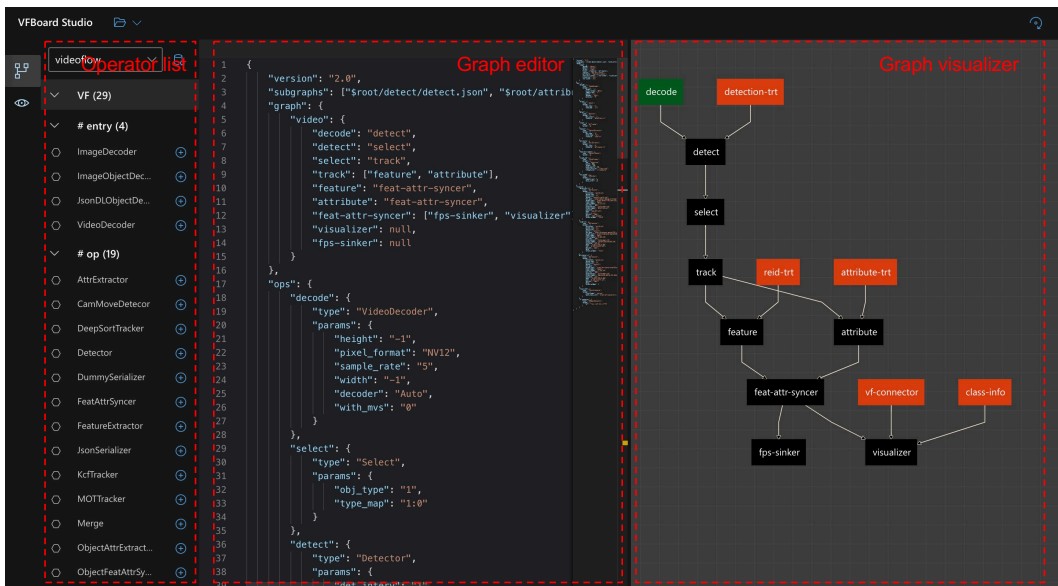

Figure 4: Computation Graph Editor & Visualizer.

## A.2 VISUALIZER

Visualizer is actually an image stream player, with abilities to draw rectangles and texts on images. It is used to visualize the real-time analysis results of algorithms, like the detected objects, object trajectories, object attributes as shown in Figure 5. It is especially useful during the algorithm development period. In production, it can be used to check if algorithms are running or if algorithms perform well. Currently it is an image stream player for better alignment of video frames and analysis results. A video player is expected from the community with the functionality to display frames together with the time aligned elements from algorithms.

## A.3 PROFILER

One of the major things in development is to profile the performance of the overall system as well as each of its components. Profiler of VideoFlow has two views: channel view and resource view. Channel view displays the running status of each video channel. As shown in Figure 6, there are 12 orders for the selected channel. The horizontal axis is the timeline. The green blocks are the executed operators as well as their execution time. The figure also verifies frame-level parallelism. The resource view currently displays the running status of deep learning models. With the horizontal axis denoting the timeline and vertical axis denoting the batch size of each inference. This can help

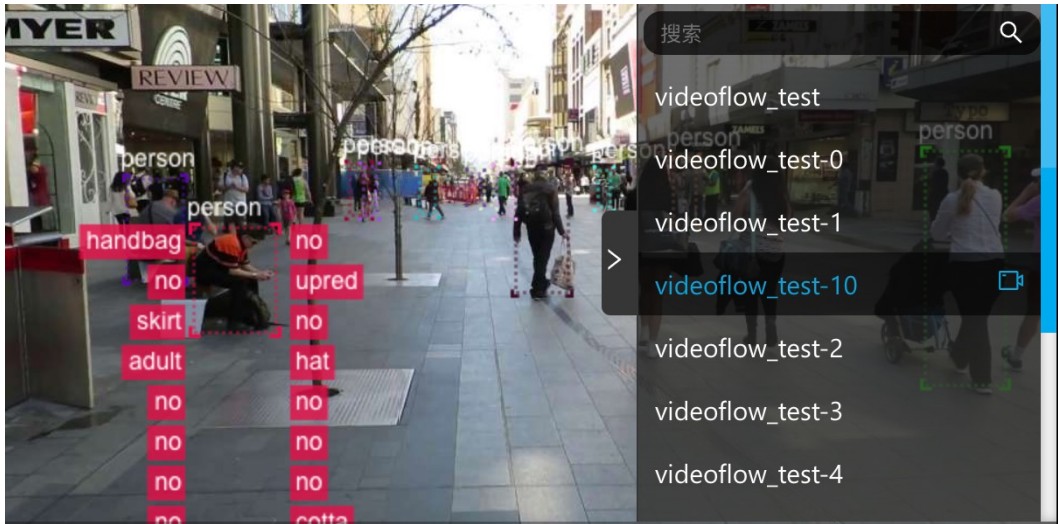

Figure 5: Real-time Analysis Visualizer.

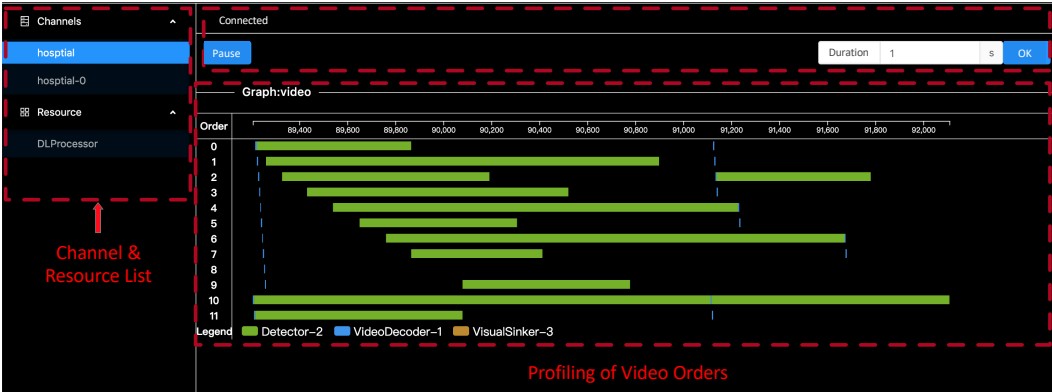

Figure 6: Performance Profiler.

to check if heterogeneous hardware resources are fully utilized or overloaded. In cases where a graph uses many deep models, it can help to analyze the bottleneck of the overall system throughout.

