# OpenReview forum: "VideoFlow: A Framework for Building Visual Analysis Pipelines"
_ICLR.cc/2021/Conference — Reject_

### Official Review · AnonReviewer2 · 2020-10-26
**A tutorial to a video analysis platform software**

**Rating:** 3
**Confidence:** 4

**Review:**

The paper presents a tutorial to a video analysis platform software, i.e., VideoFlow, which represents a video analysis task as a computation graph, provides common functions like video decoding and database storage, integrates deep learning frameworks, e.g. Caffe/Pytorch/MXNet as built-in inference engines, and supports heterogeneous hardware such as CPU/GPU/FPGA. VideoFlow also allows the customers to develop operator, decoder, and model inference extensions. The paper presents an example application of person ReID using the VideoFlow platform. The paper claims this VideoFlow software could be used in both academic and industrial scenarios.

The VideoFlow platform software is certainly a great development tool for video analysis tasks. The major concern is that if it is appropriate for ICLR to publish this tutorial which may be regarded as an endorsement to this software. Perhaps, VideoFlow could demonstrate in the exhibition of academic conferences to promote the usage in academia.

Some detailed comments:

1)	Is VideoFlow a commercial software or a free open-source software? What kind of license does VideoFlow follow? As it integrates some functions of other software like Caffe/Pytorch/MxNet.

2)	The major issue of adopting VideoFlow could be the learning curve how to use and debug applications on VideoFlow. VideoFlow appears a powerful and comprehensive video analysis platform, which abstracts and defines the interfaces among the components in a computation graph. For most of academia users, PyTorch/TensorFlow may be good to develop and verify some new deep model or algorithms. For industrial users, debugging any problem on VideoFlow requires a profound understanding of the platform. Deployment of an application could encounter issues on run-time library dependency, cross-compilation, resource limitation, e.g, on an embedded system. It would be a critical decision to adopt an open-sourced development platform for commercial applications for industrial users.

3)	Some minor questions:  As there is already a software named VideoFlow (de Armas, 2019), why not choose a different name for this software?  Vector<void*> appears a C++ implementation of a C-style concept, which implicitly requires the caller to agree with a protocol specified in other documents. The users may make mistakes that are hard to debug.

Overall, the proposed VideoFlow software is a great development tool for video analysis tasks using deep learning models. I encourage the authors to promote this software in the open-source community and demonstrate it in conference exhibition and trade shows.

---

### Official Review · AnonReviewer4 · 2020-10-29
**A fit for system communities, but not for the machine learning community**

**Rating:** 3
**Confidence:** 4

**Review:**

This manuscript describes a framework, VideoFlow, for deploying and analyzing video execution flows. VideoFlow abstracts the workload into computation graph and resource, and dynamically executes the graph given the resource.

Although this work seems helpful for system integration and deployment, I did not see a strong connection between it and the topic of this conference. In this framework, learned models are treated as an atomic node and executed on other DL libraries. The graph mentioned in this paper, as well as terms like forward/backward, has nothing to do with the graph in machine learning concept. Therefore, I do not see this paper as a good candidate for machine learning community, but a better fit for system communities.

There are some additional comments:
1. Figures should come with captions to illustrate themselves, instead of just a title.
2. Figure 2 is confusing. It is hard to infer from the figure what forward function is. The authors need to emphasize this forward/backward function is different from DL models' forward/backward.

---

### Official Review · AnonReviewer3 · 2020-10-31
**VideoFlow: A Framework for Building Visual Analysis Pipelines**

**Rating:** 4
**Confidence:** 3

**Review:**

Summary of contribution:

This paper presents VideoFlow, a domain-specific deep learning framework that focuses on building pipelines for video analytics. The goal of VideoFlow is to facilitate both development and deployment of visual data analysis pipelines for real-world applications. The paper describes the design and implementation of this framework. VideoFlow seems like a useful tool, but its novelty and contributions to the state of the art have not been clearly articulated in the paper.

Strong points:
- The need for VideoFlow is well motivated.
- Flexibility, efficiency, extensibility, and security have been identified to be the key design goals for VideoFlow's architecture.
- VideoFlow has been in practical use in the smart city domain for the past few years.
- VideoFlow supports utilizing heterogeneous hardware.

Weak points:
- The paper reads more like a system demo paper. Its contribution as a research paper is not clear.
- Good coverage of related work, but it is not clear how exactly VideoFlow compares to the state of the art. In what ways is it superior? No experimental evidence and/or user/case studies have been provided to show this.
- There is not enough experimental evidence in the paper that the four design goals are all completely met.
- It would have been nice to include more experience from the smart city deployment.

Additional comments:
- Open-sourcing VideoFlow beyond its current incubated use would help enable community's use, feedback, and contributions.

---

### Official Review · AnonReviewer1 · 2020-11-01
**An engineering introduction to a visual analysis system but not a scientific paper**

**Rating:** 3
**Confidence:** 4

**Review:**

Summary
-------

This paper introduce a framework for visual analysis (so-called VideoFlow) that will be open-sourced. VideoFlow is introduced with providing a flexible, efficient, extensible, and secure visual analysis framework for both academia and industry. It has been adopted in tens of intelligent visual analysis systems.

However, through the whole paper, the key contributions of the VideoFlow should be only counted as engineering efforts rather than any novelty in the scientific or research perspective.

Therefore, ICLR would be not an appropriate venue for the submitted paper to be published. It is more valuable for authors to submit to other venues such as OSDI (USENIX Symposium on Operating Systems Design and Implementation).

Strengths
---------

- The paper introduces the architecture of the proposed framework (VideoFlow) in detail, including operator, data flow, resource, graph construction, execution scheduling, etc. These details should be helpful for users to understand the logic of its working flow and how the framework should be used.

- The paper also presents the evaluation of person re-identification (ReID) task on VideoFlow to show its efficiency.


Weaknesses
----------

- The motivation of presenting such a practical framework is clear and understandable, however, as a research paper, scientific contributions are required to prove its novelty rather than the only engineering efforts. For such a purpose, the following points could be considered to improve the current paper towards a more scientific-like research paper.

	- Address some scientific issues that were made in the architecture of VideoFlow: original methods solving the computation/IO intensive problem, theoretical efforts for graph construction and the design of execution scheduling.

	- Provide some theoretical or mathematical guarantees of the efficiency in the working flow of execution scheduling (Fig. 2).

- The evaluation is not sufficient to show the advantages (flexible, efficient, extensible, and secure) that claimed by authors. In this paper, only the evaluation of ReID task is presented, but how about its efficiency and other advantages for other computer vision tasks such as object detection, segmentation, face recognition, and so on.

---

### Official Review · AnonReviewer5 · 2020-11-04
**Official Blind Review #5**

**Rating:** 3
**Confidence:** 4

**Review:**

This paper points out the fact that just developing models using deep learning frameworks is not the end of the story when it comes to building end-to-end visual pipelines. The paper introduces VideoFlow which is a framework that aims to improve the development process of streaming pipelines.

Compared to the widely used TensorFlow, PyTorch, and MXNet, VisualFlow focuses on more coarse-grained blocks like a "whole network" itself instead of "layers." As such, the work is developed around a rather different units of data compared to Tensors in DNNs. This work also incorporates a GUI that lets the user edit the computation graphs. I believe this work in full fledged form may help the productivity while building visual analysis applications.

Sadly, there are many shortcomings of the paper. First, the paper literally spends over 60% of the paper to describe the implementation details which does not lead to much intellectual insights. This looks more like a technical report than a research paper.

Furthermore, the paper lacks on evaluation in many aspects. For example, there is no evaluation about the potential overhead of the framework. I have listed a number of questions below with my comments.

The paper began with a luring abstract; however, after reading through the paper, the reader is left with only minor insights. Up to this point, the paper seems to be an amalgamation of various libraries and frameworks with a GUI wrapper. While I do believe the paper has some prospect as it does touch upon a real problem that does indeed take up a lot of resources in industry, I believe the paper is yet in a premature state to merit a publication at ICLR.

Also, I believe the paper would receive a more pertinent evaluation from a systems community like OSDI, SOSP, ATC. This is because these frameworks should not be just about putting things together to make something working, but should also entail a through experimentation of the performance and overhead.

Questions:
- Could you please provide running examples of applications that have graphs with a complex topology?
- How does the resource management behave for different scenarios such as underutilization? Could you show some visualization of that? For example, it would be very interesting to see how Dynamic Batching dynamically improves the utilization and the performance at runtime.
- How can this be used for on-device scenarios, or cases where there are numerous devices? For example how does this compare to NNStreamer [1]?
- If this framework were to be used in cases like Inference-as-a-Service scenario, how would this perform in terms of various QoS metrics?

[1] NNStreamer, https://nnstreamer.ai

---

### Decision · Program_Chairs · 2021-01-07
**Final Decision**

**Decision:**

Reject

**Comment:**

All reviewers appreciate the framework described in the paper and say it is a "useful tool", a "flexible, efficient, extensible, and secure visual analysis framework"  and "in full fledged form may help the productivity while building visual analysis applications."

However, the reviewers also point to significant shortcomings in terms of fit to ICLR, e.g. "looks more like a technical report than a research paper", "key contributions of the VideoFlow should be only counted as engineering efforts rather than any novelty in the scientific or research perspective", or "major concern is that if it is appropriate for ICLR to publish this tutorial which may be regarded as an endorsement to this software".

The authors did not reply to the reviewers comments.

Overall the paper does not seem to contain sufficient scientific contributions for being accepted.